# The Effect of Plasma-Activated Water Combined with Rosemary Extract (*Rosmarinus officinalis* L.) on the Physicochemical Properties of Frankfurter Sausage during Storage

**DOI:** 10.3390/foods12214022

**Published:** 2023-11-03

**Authors:** Fatemeh Zeraat Pisheh, Fereshteh Falah, Farideh Sanaei, Alireza Vasiee, Hossein Zanganeh, Farideh Tabatabaee Yazdi, Salam A. Ibrahim

**Affiliations:** 1Department of Food Science and Technology, Faculty of Agriculture, Ferdowsi University of Mashhad, Mashhad 91779-48974, Iranfereshtefalah11@gmail.com (F.F.); hossein.zanganeh64@gmail.com (H.Z.); 2Research Institute of Food Science and Technology (RIFST), Mashhad 91895-157356, Iran; ali.vasiee@gmail.com; 3Food and Nutritional Sciences Program, North Carolina Agricultural and Technical State University, E. Market Street, 1601, Greensboro, NC 24711, USA

**Keywords:** PAW, sausage, rosemary extract, physicochemical properties, shelf life

## Abstract

This study investigated the impact of plasma-activated water (PAW) and rosemary extract on the bacterial inactivation and quality attributes of Frankfurter sausages during a 6-day storage period. The antibacterial activity, total phenol content (TPC), and total flavonoid content (TFC) of the rosemary extract were evaluated. The TPC of the rosemary extract was 89.45 mg gallic acid/g dry weight, while the TFC was 102.3 mg QE/g dry weight. Even at low concentrations, the rosemary extract effectively inhibited the growth of all the tested pathogens using the Well Diffusion Agar method (WDA). The sausages were treated with different concentrations of PAW and rosemary extract and stored for 1 and 6 days. Sample B (100% rosemary extract + PAW treatment) showed the greatest reduction in microbial load and was selected for further analysis. Throughout the storage period, Sample B exhibited no significant changes in pH, moisture content, textural parameters, or sensory evaluation compared to the control group. However, the hardness and color parameters (L*, a*) of Sample B decreased, while the TBARS value increased after 6 days of storage. The combination of PAW and rosemary extract, particularly Sample B, effectively inhibited bacterial growth in the Frankfurter sausages without compromising most quality attributes. Some changes in hardness, color, and lipid oxidation were observed over the extended storage period.

## 1. Introduction

Meat and meat products are highly esteemed for their abundant protein content and essential nutrients, making them a vital part of the human diet. However, their distinctive composition, characterized by high water activity and moderate pH, renders them vulnerable to microbial growth. This susceptibility can result in spoilage and pose potential risks of foodborne illnesses [1]. To mitigate these risks, sodium nitrite and nitrate are commonly used as preservatives in sausages, such as Frankfurters, to inhibit the growth of various microorganisms, including heat-resistant spores produced by *Clostridium botulinum*, which can produce neurotoxins. These preservatives play a crucial role in extending the shelf life of sausages and improving their safety. However, these compounds have been linked to the formation of potentially carcinogenic and mutagenic nitrous compounds, particularly nitrosamines. Nitrosamines have been associated with the development of tumors in the stomach, liver, and brain, as well as damage to red blood cells. Given the potential health concerns associated with sodium nitrite and nitrate, there is a growing interest in alternative preservation methods that can ensure the safety and quality of sausages without compromising human health [2,3].

One alternative method that has been explored is the utilization of natural extracts with antimicrobial and antioxidant properties. Rosemary (*Rosmarinus officinalis* L.), a herb with a long history of medicinal and culinary use, has garnered attention for its therapeutic properties and aromatic qualities. The bioactive compounds present in rosemary, particularly the phenolic compounds, contribute to its antioxidant, antibacterial, and anti-inflammatory properties. The antioxidant properties of rosemary extracts help to inhibit or delay the cellular oxidative processes that contribute to food deterioration and microbial damage. Furthermore, rosemary extracts have demonstrated antibacterial activity against various pathogenic microorganisms. These attributes make rosemary a promising candidate as a food additive for products with a limited shelf life, as it can enhance their preservation and safety. The approval of rosemary extracts as food additives by the Food and Agriculture Organization of the United Nations (82nd JECFA-2015) further supports their use as a natural alternative to synthetic preservatives like nitrite and nitrate salts [4].

Cold plasma technology is gaining interest for various food applications [5]. PAW, short for Plasma-Activated Water, refers to water that has undergone treatment with cold plasma. This innovative technology has shown great potential as a sustainable alternative to traditional chemical reagents like chlorine, largely because of its environmentally friendly nature [6]. In recent years, PAW has gained considerable attention as a decontamination method in the food industry. Extensive studies have been carried out to assess its efficacy in treating different types of foods, and the findings have consistently shown that PAW can serve as an effective microbial decontaminant without compromising food safety and quality [7].

The aim of the present study was thus to investigate the effect of PAW and rosemary extract on the bacterial inactivation and quality attributes of sausages after 1 day and 6 days of storage.

## 2. Materials and Methods

### 2.1. Preparation of Rosemary Extraction

The dried rosemary plants were purchased at a market in Mashhad, Iran. An electric blender (Bosch Limited, Stuttgart, Germany) was used to grind the plants after they had been rinsed with running water and dried in the shade. In total, 100 g of the powdered plants was mixed with 500 mL of 70% ethanol. The mixture was stirred for 24 h at room temperature, following a method by Faria et al. (2020) [8] with some modifications; ultrasound-assisted extraction (UAE) was then performed using an ice bath. The samples were immersed in a 1000 mL beaker with ice cubes, and a titanium probe (4 mm) was used to apply focused ultrasound (Misonix, 20 KHz, 550 W, Berlin, Germany). The power was set to 40% and the extraction time was 15 min. To obtain a clear extract, the rosemary extract was centrifuged at 2990× *g* (Eppendorf 5804 R, Hamburg, Germany). The supernatants were then dried at 50 °C for 24 h and stored in a refrigerator at 4 °C. The yield of the extract was calculated as a percentage (*w*/*w*) using the following formula:Amount of extract yielded in percentage (*w*/*w*) = *w*2/*w*1 * 100(1)

*w*1 = the weight of plant powder that was macerated and treated with ultrasound (g)*w*2 = the weight of the extract obtained (g)

### 2.2. The Total Amount of Phenolic (TPC)

The TPC of each extract was determined using the Folin–Ciocalteu reagent (Sigma-Aldrich, St Louis, MO, USA). For this, 1 mL of the Folin–Ciocalteu reagent with a concentration of 1 mL (1 N) was mixed with 5.8 mL of distilled water. Subsequently, 3 mL of a 20% sodium carbonate (Na_2_-CO_3_) solution was added to the mixture. The resulting solution was then shaken in the dark at room temperature for two hours. After the incubation period, the absorbance of the solution was measured at 760 nm. To ensure accuracy and reproducibility, the tests were conducted in triplicate. A Gallic acid standard solution (Sigma-Aldrich, St Louis, MO, USA) was used to establish a standard curve. The TPC concentration was calculated in mg of gallic acid equivalent (GAE) per gram of extract, based on the equation derived from the Gallic acid standard curve [9]. The calibration formula for Gallic acid can be expressed as:Y = 0.9955X + 3.4371 (R^2^ = 1)(2)

In the equation, X represents the Gallic acid concentration (mg/L) and Y represents the corresponding absorbance value.

### 2.3. Total Flavonoid Content (TFC)

The TFC of the sample was determined using spectrophotometric methods, specifically by creating a complex of flavonoid–aluminum. A calibration curve was established using quercetin (Sigma-Aldrich, St Louis, MO, USA) as the reference substance. Initially, 10 mg of quercetin was dissolved in 80% ethanol and then diluted to concentrations of 25, 50, and 100 mg/L. Subsequently, the diluted standard solutions (0.5 mL) were added to a mixture containing 1.5 mL of 95% ethanol, 0.1 mL of 10% aluminum chloride (Sigma-Aldrich, St Louis, MO, USA), 0.1 mL of 1 M potassium acetate (Sigma-Aldrich, St Louis, MO, USA), and 2.8 mL of distilled water. The resulting mixture was incubated for 15 min at room temperature, and the absorbance was measured at a wavelength of 510 nm using a spectrophotometer [9]. The calibration formula for quercetin is as follows:Y = 0.0107X + 0.135 (R^2^ = 0.994)(3)

In this equation, X stands for the absorbance and Y for the quercetin concentration (mg/L).

### 2.4. Evaluation of Microbiological Properties

#### 2.4.1. Pathogens and Culture Conditions

The Department of Food Science and Technology at Ferdowsi University of Mashhad’s microbiological collection provided the following strains: *L. innocua* ATCC 33090, *E. coli* ATCC 25922, *P. aeruginosa* ATCC 27853, *B. cereus* ATCC 14579, *S. aureus* ATCC 25923, and *S. typhi* ATCC 6539. Before conducting the antimicrobial tests, the microbial strains were cultured for 24 h. A microbial suspension with a concentration of 0.5 McFarland standard, equivalent to 10^8^ CFU/mL of microbes, was prepared [10].

#### 2.4.2. Minimum Inhibitory Concentration (MIC)

The MIC was determined using the broth microdilution method. Serial dilutions of rosemary aqueous extract (350, 175, 87.5, 43.7, 21.8, and 10.2 mg/mL) were prepared in broth medium on a microtiter plate. Microbial suspensions with a concentration of 10^8^ CFU/mL were added to the micro-wells. Mueller Hinton Broth (MHB) from Quelab was used as the medium for the bacterial strains. The microtiter plates were then incubated at 37 °C for 24 h. The activity of the extract was assessed by observing the presence of red coloring in the wells after adding triphenyl tetrazolium chloride at a concentration of 5 mg/mL and incubation for 30 min. The MICs were determined as the lowest concentrations that completely inhibited the observable growth.

For the bacterial strains, 100 μL of culture from each well without red coloration was streaked on Mueller Hinton Agar (MHA) medium and incubated at 37 °C for 18–24 h. The lowest dilution that resulted in a complete inhibition of growth was selected as the minimum bactericidal concentration (MBC) [11,12].

#### 2.4.3. WDA Method

In the WDA method, the microbial solution was evenly spread onto Mueller Hinton Agar (MHA) medium in Petri plates using an L-shaped spreader. Subsequently, wells with a diameter of 6 mm were created on the surface of the agar, and these wells were filled with the rosemary extract (60 μL). The Petri dishes were then incubated at 37 °C for 24 h. After the incubation, the diameter of the inhibition zones around the wells, which indicates the antimicrobial activity, was measured [13,14].

### 2.5. Preparation of Sausages for Plasma Treatment

The Frankfurter sausage was prepared in the laboratory of Ferdowsi University of Mashhad, Iran, under conditions similar to those of commercial production. The sausages were made using the same recipe, but with different amounts of rosemary extract and nitrate. Two groups of sausages were produced to investigate the effects of combining PAW with rosemary extract on the microbiological and physicochemical properties of the sausages during a six-day storage period (refer to Table 1). The first group, without PAW treatment, served as a comparison to the second group, which received PAW treatment. To make the sausages, minced beef was combined with the following ingredients: meat (55%) 110 g, oil (12%) 24 g, ice (21%) 42 g, salt (1.5%) 3 g, spices (2%) 4 g, and starch (8.15%) 16.3 g. As indicated in Table 1, rosemary extract and nitrate were added to the sausages after combining the meat with the primary ingredients. The sausages were cooked in a water bath (MaXturdy 45, Daihan Scientific, Wonju, Republic of Korea) until reaching a temperature of 75 °C. After cooking, the products were rapidly chilled in an ice slurry for 20 min and then stored in the dark at a temperature from 2 °C to 3 °C for analysis and cold plasma treatment [2].

### 2.6. Cold Plasma Treatment

In this study, the PAW was prepared using a gliding arc discharge plasma jet operating at atmospheric pressure. The plasma jet nozzle was positioned 5 mm above the water surface. The electrodes were supplied with electrical power from a high-voltage generator with a voltage of 5 kV and a frequency of 40 kHz. The input power was set at 750 W. Compressed air at approximately 0.18 MPa was used as the working gas, with a flow rate of 30 L/min at the jet outlet. To obtain the PAW, 50 mL of sterile distilled water was exposed to plasma activation for 15 min. The sausage samples were divided into equal-sized pieces weighing 12 g. Subsequently, each sausage sample was individually placed in a sterile container with 50 mL of PAW at room temperature for 15 min before being stored. Microbial tests, specifically the total plate counts, were conducted on the 1st and 6th days of storage. Physicochemical tests were performed on the treatment selected from the first and second groups based on the reduction in microorganisms within 6 days after treatment. The microbial load reduction (according to the logarithmic cycle) was calculated using the following formula [15]:Reduction cycle = Log N0/N(4)

N: the number of colonies in the treatment sampleN0: the number of colonies in the control sample

### 2.7. Microbial Analysis

To homogenize 12 g of Frankfurter-type sausage for 3 min, 10^8^ mL of 0.1% peptone water was used in a sterile bag Stomacher (Combourg, France). The same diluent was also used to create serial dilutions. After 48 h of growth at 37 °C using plate count agar (PCA), the total number of plate counts was determined. The microbial counts were assessed on days 1 and 6 of storage [2,16].

### 2.8. pH Value

We utilized a Model 700S Waring^®^ Blender (Waring^®^ Products, Torrington, CT, USA) to homogenize 10 g of sausage in 100 mL of distilled water for 30 s. The pH measurements were carried out using a pH meter from Metrohm (Herisau, Switzerland). The pH measurements were conducted in triplicate for each treatment on the 1st and 6th days of storage for both the optimized sample and the control sample [16].

### 2.9. Moisture Content

The samples were subjected to temperature conditions of 100–102 °C for a moisture analysis. Two different values were obtained for each sample. The moisture content was determined on the 1st and 6th days of storage for both the optimized sample and the control sample [17].

### 2.10. TBARS Value

TBARS (Thiobarbituric Acid Reactive Substances) were measured using a spectrophotometer (UV 2100, Unico Scientific Instruments, Dayton, NJ, USA) as an indicator of lipid oxidation after 6 days of storage. To prepare the samples, 10 g of Frankfurter-style sausage was homogenized with 25 mL of 20% trichloroacetic acid (TCA) and 20 mL of distilled water for 30 s using a blender. The resulting mixture was then centrifuged at 1000× *g* for 20 min, and the supernatant was filtered through Whatman No. 1 filter paper. In a test tube, 2 mL of 0.02 M aqueous 2-thiobarbituric acid was added to 2 mL of the filtrate. The tube was then heated in a bath of boiling water for 20 min and subsequently cooled in tap water for 5 min. The absorbance of the sausage was measured at 532 nm, and the amount of malondialdehyde was determined per kg of sausage as an indicator of lipid oxidation [2].

### 2.11. Color

A Hunter Lab colorimeter (Color Flex 45/0, Model Colorimeter, Reston, VA, USA) was employed to measure colors using a D65 illuminant. The colorimeter was calibrated using white and black standard plates, resulting in L* = 93.01, a* = 41.11, and b* = 1.30 values. The colorimeter readings were obtained from the surface of the samples. Six readings were taken for each sample and then averaged [18].

### 2.12. Texture Profile Analysis

A texture profile analysis (TPA) was conducted at room temperature on the samples (sample B and control) after 6 days of storage using a TA.XTA2i texture analyzer (Texture Technologies Corp., Scarsdale, NY, USA/Stable MicroSystems, Godalming, UK). The Texture Expert Exceeds software was utilized for the data analysis. A cylindrical probe TA25/1000 was employed to measure the texture, specifically the force compression (g). Prior to testing, the instrument was calibrated using a 40 kg load cell. The test speed was set at 1 mm/s, and the probe was allowed to compress the sample by 0.4 mm. The parameters assessed for the texture evaluation included Hardness, Cohesiveness, and Springiness [19].

### 2.13. Sensory Evaluation

On the first day of storage, a group of 12 experienced panelists, consisting of students and academic staff from the Department of Food Science and Technology at Ferdowsi University of Mashhad, assessed the sensory quality of the sausages treated with rosemary extracts and PAW (sample B) compared to the control group (sample G, untreated with PAW, 100% nitrate). A five-point hedonic scale was utilized to rate the products, ranging from 1 (strong dislike) to 5 (strong like), with 3 indicating neither like nor dislike for flavor, color, and tenderness. The overall acceptability was calculated as the sum of scores for each attribute. The evaluations were conducted in a well-lit room [2].

### 2.14. Statistical Analysis

The experimental data were analyzed using a one-way analysis of variance (ANOVA). The General Linear Model in SPSS (Version22.0; IBM Corp., Armonk, NY, USA) was utilized for this analysis. To determine significant differences between the mean values, Duncan’s test was conducted at a significance level of *p* < 0.05. The experiments were independently conducted three times [20].

## 3. Results and Discussion

### 3.1. Phytochemical Properties of Rosemary Extract

Several studies have demonstrated the diverse biological activities of rosemary extracts, including hepatoprotective, antifungal, antioxidant, and antibacterial properties [21]. These qualities are primarily attributed to the presence of phenolic acids, phenolic diterpenes, and flavonoids in rosemary, with a particular emphasis on its antioxidant properties [22]. In the present study, the TPC and TFC of the rosemary extract were approximately 89.45 mg gallic acid/g dr.wt and 102.3 mg QE/g dr.wt, respectively. In a study conducted by Pereira et al. (2017) [23], the TFC of rosemary was found to be 24.26 mg QE/g dr.wt, while the TPC was 40.91 mg gallic acid/g dr.wt. Rashidaie Abandansarie et al. (2019) reported a TPC ranging from 434.11 to 699.13 μg gallic acid/g dr.wt in their study on rosemary extracts [24]. The variations in the phenolic compound levels observed in our study may be attributed to the environmental and genetic characteristics and harvesting conditions of the rosemary, as well as differences in the extraction process and choice of solvents [25].

### 3.2. The Evaluation of Rosemary Extract’s Antibacterial Activity

According to Table 2, the MICs of the rosemary extract against *S. aureus*, *L. innocua*, *P. aeruginosa*, and *E. coli* were 43.7, 87.5, 43.7, and 43.7 mg/mL, respectively. The MBCs of the rosemary extract against *S. aureus*, *P. aeruginosa*, and *E. coli* were 87.5 mg/mL, while it was 175 mg/mL against *L. innocua*.

In the WDA techniques (Table 3), the rosemary extract at the lowest concentration exhibited an inhibitory zone against all tested microorganisms. Conversely, the WDA technique revealed that, at the highest concentration of the extract, *E. coli* showed the largest inhibitory zone, while *S. aureus* exhibited the smallest inhibitory zone. Rosemary extract has demonstrated potent antibacterial activity against both Gram-negative and Gram-positive bacteria. This can be attributed to the presence of carnosic acid, which serves as the primary antimicrobial component of the extract. Carnosic acid interacts with the cell membrane, resulting in genetic material changes, alterations in nutrition and electron transport, the leakage of cellular components, and the inhibition of fatty acid synthesis [25]. Studies by Ekambaram et al. (2016) showed that rosemary extracts are effective in inhibiting the growth of *S. aureus* [26]. It is possible that rosmarinic acid, a component of rosemary, acts as an antibacterial agent by affecting the virulence factors and surface proteins of *S. aureus*. 1,8-cineol disrupts the cell structure and induces the release of extracellular substances, functioning as an antibacterial agent. Additionally, the effectiveness of rosemary against *E. coli* is attributed to the synergistic impact of the minor constituents in the volatile fraction.

### 3.3. Microbial Analysis

The effects of nitrate, rosemary extract, and PAW treatment on the total plate count in the sausages during storage for 1–6 days are presented in Figure 1. Samples containing nitrate and rosemary extract (treated with PAW and samples with no treatment) did not show any significant statistical differences on the first day of storage. Sample E (treated with 75.0% nitrate + 25% rosemary extract and not treated with PAW) had the lowest microbial load. All the samples, except for samples B (treated with 100% rosemary extract and PAW), H (treated with 100% nitrate and PAW), and C (treated with 50% nitrate and 50% rosemary extract, without PAW treatment), showed an increase in microbial load after the 6-day storage period. However, compared to the microbial load on the first day of storage, this increase was not significant. Samples B, H, and C had a significantly lower microbial load on the sixth day of storage compared to the other samples. Sample B, which exhibited a decrease of 0.09 log CFU/g during the storage period compared to the other samples, was chosen for PAW treatment. PAW is produced by the reaction of non-thermal atmospheric plasma with water and contains an abundance of highly reactive oxygen (ROS) and nitrogen (RNS) species [27]. There is significant disagreement regarding the reactive species responsible for deactivating microorganisms within a wide range of PAW. According to many authors, the antibacterial activity of PAW is attributed to the synergistic effects of its physicochemical properties, including ROS, RNS, pH, and UV radiation. Additionally, carnosic acid, carnosol, and rosmarinic acid are the main bioactive antimicrobial components found in rosemary extracts [6,28]. The significant reduction in the microbial load in sample B was likely due to the combined effect of rosemary extract and plasma processing in this investigation. In this regard, sample B and sample G as a control (without treatment with PAW, 100% nitrite) were selected to investigate and compare the simultaneous effect of PAW and rosemary extract on the physicochemical characteristics of the sausage samples during storage.

### 3.4. pH Value and Moisture Content

Table 4 presents the pH values and moisture contents of the sausages during storage at 4 °C. By the end of the storage period, the pH levels of the control (samples G) and optimized sample (sample B) had decreased, although this change was not statistically significant. At the beginning and end of the storage, sample B (100% extract) exhibited slightly lower pH levels compared to the control sample. Several factors, such as lipolysis, which releases free fatty acids from meat [16], have been implicated in the decrease in pH values. As shown in Table 4, the addition of rosemary extract and PAW treatment had no effect on the moisture content. After storage, the moisture content of all the samples (control and B) experienced a decrease, although this change was not statistically significant. Six days after storage, the moisture content of the sausages (sample B) ranged from 59.14% to 58.12% (Table 4). No significant differences were observed between the optimized samples and the control samples.

### 3.5. Color

According to Table 4, the CIE L* and CIE a* values of sample B (100% rosemary extract) were lower than those of the control sample (sample G) after treatment with PAW and 6 days of storage (*p* > 0.05). However, there was no statistical difference between the b* values of the optimized sample and the control sample after storage. Previous studies have shown that the addition of rosemary extract can significantly reduce the L* values of chicken breast [29] and fresh chicken sausage [30]. This color change may be attributed to the darker color of the rosemary extract and a browning process that occurs in the extract due to the conversion of phenolic compounds into quinones [29,31]. PAW treatment also affects the redness (a* value) of meat products. Similar to our findings, the a* values of plasma-treated bresaola [32], orcine longissimus dorsi muscle samples [33], and pork loins [34] decreased significantly with an increasing storage time. The color changes induced by PAW treatment in our study may be attributed to ROS and RON species generated during plasma activation [35,36,37]. In a study by Schilling et al. (2018) [38], it was found that the antioxidant properties of rosemary extract can effectively protect myoglobin from oxidation, but it needs to pass through the cell membrane due to the intracellular position of the pigment. The rosemary extract may have prevented the formation of myoglobin, which is responsible for the brown color of meat, by inhibiting the development of the primary and secondary lipid oxidation products that accelerate myoglobin oxidation [16].

### 3.6. TBARS Value

After the PAW treatment and 6 days of storage, the TBARS value of sample B was higher than that of the control sample (sample G) (Figure 2).

However, the increase observed in this study was relatively minor, with a value of 0.5 MDA/kg. It is important to note that PAW contains reactive species that can react with the fatty acids present in meat. This reaction can result in the formation of malondialdehyde (MDA) through the degradation of polyunsaturated fatty acids. The presence of MDA can contribute to the development of unpleasant odors and rancid flavors in meat. Similar to our findings, Qian et al. (2022) [39] demonstrated that PAW can accelerate the lipid oxidation in chicken meat due to the peroxides generated during treatment. Additionally, Luo et al. (2019) [40] found that the TBARS value of dried pork loins decreased significantly (from 0.41 mg/kg to 0.22–0.33 mg/kg) when cured with PAW compared to the control sample. The rate of lipid oxidation in food products is influenced by various factors, including the composition of the food and the conditions of the plasma treatment [41]. Gao et al. (2019) also showed that rosemary can reduce TBARS levels in cold plasma-treated chicken breast patties after 5 days of storage [29]. Therefore, the presence of rosemary extract in sample B after PAW treatment and 6 days of storage can be considered as both a reason and an impediment to the further increase in TBARS levels.

### 3.7. Texture Parameter of Sausages

Table 5 presents the texture profile of sample B (100% rosemary extract) after a 15 min PAW treatment and 6 days of storage. The hardness parameters of the sausage sample exhibited a significant change following the 15 min PAW treatment. It was observed that the hardness values of the sausage sample were significantly lower than those of the control sample (sample G) (*p* < 0.05). Consistent with our results, the findings of Zhao et al. (2018) [42] demonstrated that the hardness of fresh beef treated with PAW was significantly lower than that of other treatments after 12 days of storage. This decrease in hardness was likely a result of the oxidative degradation of proteins by PAW during the 6-day storage period. Similarly, the addition of rosemary powder, as reported by Jung et al. (2015) [43], contributed to a reduced dryness in the sausage, allowing for longer retention of water content. Based on our study results, the decrease in hardness observed in the optimized sausage samples after PAW treatment can be attributed to the combined effect of rosemary extract and PAW. However, in our study, there was no statistically significant difference between the cohesion and elasticity values of the PAW-treated samples and the control samples.

### 3.8. Sensory Evaluation

The evaluation of PAW-treated foods traditionally relies on sensory characteristics such as color, appearance, aroma, and taste, which significantly influence the perceived quality and consumer acceptance of these food products post-PAW treatment [44]. Figure 3 illustrates the sensory characteristics of sample B (100% rosemary extract) after a 15 min PAW treatment compared to the control (sample G). In our study, the taste, tenderness, and overall acceptability of the sausage did not undergo significant changes with PAW treatment compared to the untreated samples. However, the color of the sausage was significantly reduced by the PAW treatment (*p* < 0.05). Jongberg et al. (2013) [45] discovered that the addition of plant extracts (rosemary and green tea) altered the color of sausages, particularly after 4 weeks of storage. The sausages, which were prepared without nitrite, exhibited a grayer appearance compared to conventionally salted sausages with nitrite. Furthermore, PAW treatment has resulted in noticeable color changes in various foods, particularly meat [43,46]. The product type, PAW treatment parameters, and storage conditions were identified as the influential factors in color alterations [44]. Liao et al. (2020) [47] conducted a study to investigate the impact of PAW treatment on the quality of fresh beef. Their research findings indicated that the utilization of PAW treatment did not adversely affect the quality of fresh beef compared to the control group. The changes in sensory acceptance attributed to PAW can be attributed to the production of ROS and reactive RNS during plasma discharge [48].

## 4. Conclusions

In the present study, the application of PAW technology in conjunction with other technologies effectively eliminated microorganisms from Frankfurter sausage while minimizing significant alterations to the physical and chemical properties of the sausage. This result highlights the importance of selecting the appropriate treatment conditions to ensure the preservation of the sausage using PAW. In addition, the results suggest the need for additional research to investigate the combined effects of different preservation technologies to enhance the effectiveness of PAW while minimizing any negative impact on taste, nutritional value, and functionality. A comprehensive analysis of the reactive species in PAW is also necessary in order to optimize the treatment conditions for specific food products and to improve the targeted application of reactive species.

## Figures and Tables

**Figure 1 foods-12-04022-f001:**
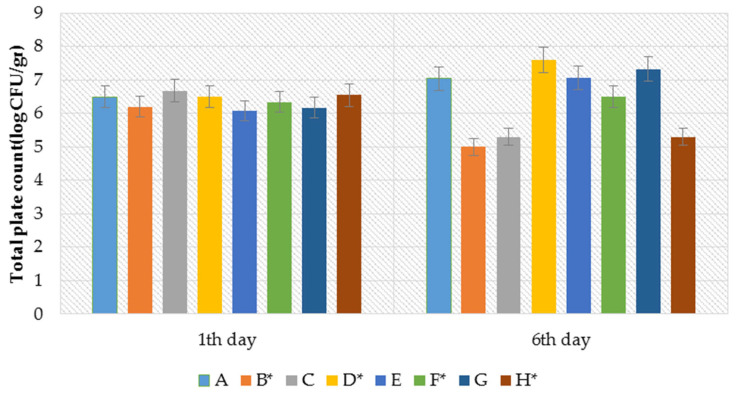
Microbial reduction (log) in sausages with PAW (15 min) treatment and without treatment of PAW during storage (1, 6), (*p* ≤ 0.05). * sausage without PAW treatment.

**Figure 2 foods-12-04022-f002:**
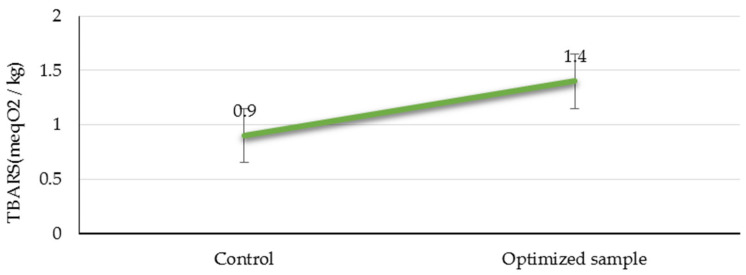
The TBARS value in the optimized sample (sample B) following CP treatment and the end of 6-day storage time.

**Figure 3 foods-12-04022-f003:**
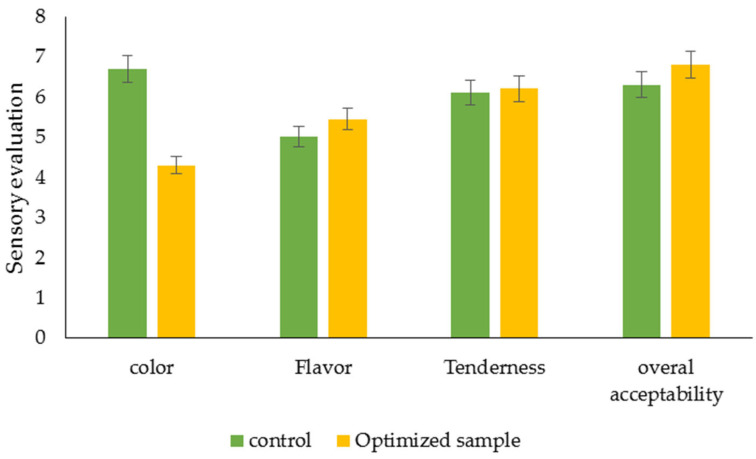
The effect of PAW treatments for 15 min on the sensory properties of an optimized sample (sample B) in 1 day of storage (*p* ≤ 0.05).

**Table 1 foods-12-04022-t001:** Groups of sausages with varying amounts of rosemary extract and nitrate.

Treatment	Nitrate (%)	Rosemary Extract (%)
A *	0	100
B	0	100
C *	50	50
D	50	50
E *	75	25
F	75	25
G *	100	0
H	100	0

* without PAW treatment.

**Table 2 foods-12-04022-t002:** MIC and MBC of extract on pathogenic bacteria.

Microorganisms	MIC (mg/mL)	MBC (mg/mL)
*Staphylococcus aureus*	43.7	87.5
*Listeria innocua*	87.5	175
*Pseudomonas aeruginosa*	43.7	87.5
*Escherichia coli*	43.7	87.5

**Table 3 foods-12-04022-t003:** Average inhibition zone (mm) of extract concentrations on pathogenic bacteria (WDA methods).

Zone of Inhibition (mm)				
Microorganisms	Well Diffusion Agar			
	Concentrations (mg/mL)			
	50	100	200	400
*Pseudomonas aeruginosa*	12.15 ± 0.52 ^a^	15.24 ± 0.12 ^b^	18.45 ± 0.42 ^a^	20.25 ± 0.31 ^b^
*Escherichia coli*	14.31 ± 0.30 ^c^	17.10 ± 0.13 ^d^	20.10 ± 0.19 ^b^	22.08 ± 0.08 ^c^
*Staphylococcus aureus*	12.31 ± 0.31 ^a^	16.10 ± 0.15 ^c^	19.10 ± 0.11 ^c^	19.10 ± 0.12 ^a^
*Listeria innocua*	13.31 ± 0.20 ^b^	14.10 ± 0.03 ^a^	20.10 ± 0.12 ^b^	20.12 ± 0.02 ^b^

^a–c^ Means with different subscripts within a column significantly different at *p* < 0.05.

**Table 4 foods-12-04022-t004:** Change in pH, moisture content, and color values of optimized sausages after PAW treatment during storage at 4 °C.

Analysis	Treatment	1st Day	6th Day
pH	Control	6.59 ± 0.03 ^a^	6.09 ± 0.02 ^a^
	Optimized sample	6.32 ± 0.05 ^b^	6.12 ± 0.04 ^a^
Moisture content (%)	Control	62.14 ± 0.4 ^a^	58.21 ± 0.22 ^a^
	Optimized sample	59.14 ± 0.42 ^b^	58.12 ± 0.12 ^a^
L*	Control	51.94 ± 0.2 ^a^	49.34 ± 0.12 ^a^
	Optimized sample	47.04 ± 0.25 ^b^	47.44 ± 0.11 ^b^
a*	Control	14.91 ± 0.19 ^a^	12.91 ± 0.01 ^a^
	Optimized sample	4.21 ± 0.11 ^b^	0.01 ± 0.01 ^b^
b*	Control	16.91 ± 0.21 ^a^	15.01 ± 0.29 ^a^
	Optimized sample	17.29 ± 0.14 ^b^	16.12 ± 0.01 ^b^

^a,b^ Means with different subscripts within a column significantly different at *p* < 0.05. Control and optimized sample are sample G and sample B, respectively.

**Table 5 foods-12-04022-t005:** The texture profile of the optimize sample (100% rosemary extract) following PAW treatment (15 min) and end of 6 days of storage.

Analysis	Hardness (g)	Springiness	Cohesiveness (%)
Control (sample G)	2995.33 ± 48.65 ^a^	0.89 ± 0.02 ^a^	0.50 ± 0.04 ^a^
Optimized sample (sample B)	2014.23 ± 52.15 ^b^	0.91 ± 0.01 ^a^	0.49 ± 0.01 ^a^

^a,b^ Means with different subscripts within a column significantly different at *p* < 0.05.

## Data Availability

The data presented in this study are included in the article.

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
