# Peer review of "The Effect of Plasma-Activated Water Combined with Rosemary Extract (Rosmarinus officinalis L.) on the Physicochemical Properties of Frankfurter Sausage during Storage"

_foods, 2023, doi:10.3390/foods12214022_

Round 1
Reviewer 1 Report
Comments and Suggestions for Authors
The article is interesting and deals with the application of the non-thermal plasma method, specifically plasma-activated water in combination with rosemary extracts on sausages.
The paper contains a number of errors, sloppiness and carelessness, comments and possibilities for improvement are attached below:
- The wording is sometimes confusing, it is recommended to check this in detail and is probably due to errors in translation.
- The authors make recurrent use of "we". It would be advisable not to use this form.
In L14 the WDA method is mentioned, as it is the first time it is used in the document, the full name could be used and from then onwards the abbreviation could be used.
It is necessary to standardize the treatment with rosemary, for example in the summary they talk about % but in the rest of the document they talk about concentrations.
In l16 it talks about the best sample, it should not say the best treatment.
In l28 it is mentioned that nitrites and nitrates are related to preservation, but their contribution to the development of typical sausage characteristics such as colour is left out, revise and reword.
Check the names of the micro-organisms, not all are written in italics and some are abbreviated.
L31 change Nitrosamines to nitrosamines.
Revise the wording of L35, it would be better to talk about natural extracts instead of plants.
In l43 it is mentioned that fao and EFSA approved its use in food, it would be interesting to indicate the date from which it happens.
l45-46 is disconnected from the rest, improve the wording.
Revise the wording in l48 it still does not establish a relationship with thermal and non-thermal methods.
L56 revise the wording when talking about quality parameters.
In l58 there is a need to mention something about what the negative effects of PAW would be.
Are there any quality categories or classifications for rosemate?
In l71 it mentions information about a computer, shouldn't this go after blender...?
In l72 it talks about laboratory temperature, shouldn't it be room temperature?
Check in l77 about the extraction, it is not understood.
In l79 it mentions filtration, what type of filter was used?
In l80 there is a mention of refrigerated storage, at what temperature was this done?
Check the subindices and superindices throughout the document.
In 2.3 there is no mention of the equipment and conditions used.
Review and standardize on concentrations of rosemary extract L116.
l117 108 CFU or 10E8?
In l135 a national standard is mentioned, this is only locally valid, evaluate if this information is really needed.
In 2.6 it is necessary to mention complete information about the application of PAW like e.g. frequency, gas characteristics, type of plasma generator. It is also mentioned that the samples were cut, in this respect it is not clear whether the cut samples had a weight of 25g.
It is also mentioned that a treatment was optimized, however, there is no information about this optimization and what were the parameters considered for this.
In l169 it talks about two links, it is not clear what is meant by that, also it mentions treatment in duplicate, but in other parts of the document it talks about triplicate?
In l173 it says that the samples were dried. Wouldn't it be better to say that the samples were subjected to temperatures of 100-102 for moisture analysis?
In l190 there is no information about the server for the colour measurements. It is not clear why it was necessary to wrap the samples in film.
In l98 the equation for calculating the total colour difference is presented, however, this is not used in the paper, please check.
In l206 a very old reference is mentioned, please revise and update.
In 2.13 From what is reported it is not clear exactly how the sensory analysis was carried out.
In 2.14 it could be expanded.
In some parts of the document the font has different sizes, revise and standardize.
In general terms the discussion is limited to presenting the results but there is no deeper analysis of the related phenomena.
In l277 they talk about the abundance of ros and rons, did they measure it ? there are kits that allow to measure in addition to this the generation of ozone.
In l298 what is the contribution about gram + and LAB, it looks out of context.
In l300 what could be the reason for affecting the humidity of the samples, more information is needed on how the PAW was applied and the subsequent handling of the samples.
In l317 it is the water that evaporates, not the moisture, please check.
The discussion on sensory evaluation can be improved, its input is very limited and needs to be expanded.
The conclusion should be rewritten in terms of generating new knowledge and possible applications.
In table 1 replace treatment by Treatment
In table 4 and 5 place heading for the first column (analysis/mediation).
Revise the texture analysis, it is not clear how springiness and cohesiveness were measured.
What is the meaning of drawing a straight line between two control points and optimize sample?
Improve Figure 3 and add error bars.
Comments on the Quality of English Language
The wording could be improved, as it contains many errors and is sometimes confusing.
Author Response
-The actions and responses carried out based on the comments made by the reviewers are listed below and the modified sections of the manuscript text are highlighted in Red.
-Responses to Reviewer1:
- The wording is sometimes confusing, it is recommended to check this in detail and is probably due to errors in translation.
Answer: it’s corrected.
- The authors make recurrent use of "we". It would be advisable not to use this form.
Answer: Thanks, it’s corrected.
In L14 the WDA method is mentioned, as it is the first time it is used in the document, the full name could be used and from then onwards the abbreviation could be used.
Answer: Thanks, it’s corrected.
It is necessary to standardize the treatment with rosemary, for example in the summary they talk about % but in the rest of the document they talk about concentrations.
Answer: Thanks, it’s corrected.
In l16 it talks about the best sample, it should not say the best treatment.
Answer: Thanks, it’s corrected.
In l28 it is mentioned that nitrites and nitrates are related to preservation, but their contribution to the development of typical sausage characteristics such as colour is left out, revise and reword.
Answer: Thanks, it’s corrected.
Check the names of the micro-organisms, not all are written in italics and some are abbreviated.
Answer: Thanks, it’s corrected.
L31 change Nitrosamines to nitrosamines.
Answer: Thanks, it’s corrected.
Revise the wording of L35, it would be better to talk about natural extracts instead of plants.
Answer: Thanks, it’s corrected.
In l43 it is mentioned that fao and EFSA approved its use in food, it would be interesting to indicate the date from which it happens.
Answer: Thanks, it’s corrected.
l45-46 is disconnected from the rest, improve the wording.
Answer: it’s corrected.
Revise the wording in l48 it still does not establish a relationship with thermal and non-thermal methods.
Answer: it’s removed.
L56 revise the wording when talking about quality parameters.
Answer: it’s revised.
In l58 there is a need to mention something about what the negative effects of PAW would be.
Answer: it’s removed.
Are there any quality categories or classifications for rosemate?
Answer : yes, Rosmarinus officinalis L. (family Lamiaceae; common name rosemary)
In l71 it mentions information about a computer, shouldn't this go after blender...?
Answer: Thanks, it’s corrected.
In l72 it talks about laboratory temperature, shouldn't it be room temperature?
Answer: Thanks, it’s corrected.
Check in l77 about the extraction, it is not understood.
Answer: Thanks, it’s corrected.
In l79 it mentions filtration, what type of filter was used?
Answer: It’s revised.
In l80 there is a mention of refrigerated storage, at what temperature was this done?
Answer:4c
Check the subindices and superindices throughout the document.
Answer: Its done.
In 2.3 there is no mention of the equipment and conditions used.
Answer: its corrected.
Review and standardize on concentrations of rosemary extract L116.
Answer : The concentrations of rosemary extract used in the minimum inhibitory concentration method(MIC) to investigate the antimicrobial effect of rosemary extract against pathogens are different from the values used in the sausage formulation.
l117 108 CFU or 10E8?
Answer: 108 Cfu
In l135 a national standard is mentioned, this is only locally valid, evaluate if this information is really needed.
Answer: Thanks, its removed.
In 2.6 it is necessary to mention complete information about the application of PAW like e.g. frequency, gas characteristics, type of plasma generator. It is also mentioned that the samples were cut, in this respect it is not clear whether the cut samples had a weight of 25g.
Answer: Thanks, its corrected.
It is also mentioned that a treatment was optimized, however, there is no information about this optimization and what were the parameters considered for this.
Answer: based on the reduction of microorganisms within 6 days after treatment. To examine the impact of PAW and rosemary extract on the microbiological and physicochemical features of sausages over the six-day storage period, two groups of sausages were produced (Table 1). The first group (without plasma treatment) was used to compare to the second group (with plasma treatment).after treatment, Microbial tests(total plate count) were performed on the 1st and 6th days of storage, followed by physicochemical tests on the treatment chosen from the first and second groups based on the reduction of microorganisms within 6 days after treatment.
In l169 it talks about two links, it is not clear what is meant by that, also it mentions treatment in duplicate, but in other parts of the document it talks about triplicate?
Answer: Thanks, its corrected.
In l173 it says that the samples were dried. Wouldn't it be better to say that the samples were subjected to temperatures of 100-102 for moisture analysis?
Answer: Thanks, its corrected.
In l190 there is no information about the server for the colour measurements. It is not clear why it was necessary to wrap the samples in film.
Answer: its corrected.
In l98 the equation for calculating the total colour difference is presented, however, this is not used in the paper, please check.
Answer: Thanks, its corrected.
In l206 a very old reference is mentioned, please revise and update.
Answer: Thanks, its corrected.
In 2.13 From what is reported it is not clear exactly how the sensory analysis was carried out.
Answer: its corrected.
In 2.14 it could be expanded.
In some parts of the document the font has different sizes, revise and standardize.
Answer: its corrected.
In general terms the discussion is limited to presenting the results but there is no deeper analysis of the related phenomena.
Answer: for this section we need more time.
In l277 they talk about the abundance of ros and rons, did they measure it ? there are kits that allow to measure in addition to this the generation of ozone.
Answer: Zhao et al(2018), measured it.
In l298 what is the contribution about gram + and LAB, it looks out of context.
Answer: its corrected.
In l300 what could be the reason for affecting the humidity of the samples, more information is needed on how the PAW was applied and the subsequent handling of the samples.
Answer: In this research, a gliding arc discharge plasma jet was used to prepare PAW under atmospheric pressure. The plasma jet nozzle was positioned 5 mm above the water surface. The electrodes were supplied with electrical power from a high voltage generator with a voltage of 5 kV and frequency of 40 kHz. The input power was set at 750 W. Compressed air at approximately 0.18 MPa was used as the working gas, with a flow rate of 30 L/min at the jet outlet. To obtain PAW, 50 mL of sterile distilled water was exposed to plasma activation for 15 minutes. The samples were divided into equal-sized pieces (25g). Finally, the sausage samples were placed individually in sterile containers with 50 mL of PAW at room temperature for 15 min before being stored. Microbial tests(total plate count) were performed on the 1st and 6th days of storage, followed by physicochemical tests on the treatment chosen from the first and second groups based on the reduction of microorganisms within 6 days after treatment.
In l317 it is the water that evaporates, not the moisture, please check.
Answer: its corrected.
The discussion on sensory evaluation can be improved, its input is very limited and needs to be expanded.
Answer: its corrected.
The conclusion should be rewritten in terms of generating new knowledge and possible applications.
Answer: its corrected.
In table 1 replace treatment by Treatment
Answer: its corrected.
In table 4 and 5 place heading for the first column (analysis/mediation).
Answer: its corrected.
Revise the texture analysis, it is not clear how springiness and cohesiveness were measured.
Answer: its corrected.
What is the meaning of drawing a straight line between two control points and optimize sample?
Answer: its corrected.
Improve Figure 3 and add error bars.
Answer: its corrected.
Reviewer 2 Report
Comments and Suggestions for Authors
The manuscript titled “Investigation effect of plasma-activated water combined with rosemary extract (Rosmarinus officinalis, L.) on physicochemical properties of Frankfurter sausage during storage” investigates the influence of synergistic effect of rosemary extract and plasma-activated water on the quality of Frankfurter sausage. The topic of article is interesting and up-to-date. However, there are issues that need to be clarified. Therefore, the paper needs Major Revision.
11. Lines 59-66: It is enough to present this information in Material and methods section. Please present the aim of the research instead.
22. Lines 87-88: Please correct the chemical formula.
33. Lines 90-93: What do you mean by typical gallic acid graph? Usually you prepare calibration curve. What is more the method description should be more precise.
44. Line 139: Please be more precise. You just sprayed the sausage with plasma treated water.
55. Point 2.5: Why did you not use plasma treated water in the recipe? The method you used will not find the practical application in the industry.
66. Line 185-186: The sentence needs to rephrased.
77. Point 2.13: What did you examine: sensory quality or acceptability? The more detailed information on the panel training is needed in case of sensory quality.
88. Point 2.14: The statistical analysis description needs to be elaborated.
99. Line 227: Full stop after “weight” is not needed.
110. Line 229 and 231: Please use the proper way to present the units.
111. Line 234 and elsewhere: Please use the same size of the font.
112. Lines 316-317: What does it have to do with your research?
113. Line 335-339: Was the research material prepared similarly to yours? If so please be more precise.
114. Line 361: What was the control sample?
115. General remarks: The influence of a rosemary extract on the quality of meat products is well known. What is interesting is comparison of samples with added plasma activated water and those without this addition. However, you have resigned from the comparison after microbiological tests. Therefore the precise aim of the research is needed to assess its soundness.
Comments on the Quality of English LanguageGenerally the English quality is good. However, several sentences need to be rephrased as indicated above.
Author Response
-The actions and responses carried out based on the comments made by the reviewers are listed below and the modified sections of the manuscript text are highlighted in Red.
-Responses to Reviewer2:
- Lines 59-66: It is enough to present this information in Material and methods section. Please present the aim of the research instead.
Answer: Thanks, it’s corrected.
- Lines 87-88: Please correct the chemical formula.
Answer: Thanks, it’s corrected.
- Lines 90-93: What do you mean by typical gallic acid graph? Usually you prepare calibration curve. What is more the method description should be more precise.
Answer: Thanks, it’s corrected.
- Line 139: Please be more precise. You just sprayed the sausage with plasma treated water.
Answer: This section was corrected. The translation was wrong and incomplete.
- Point 2.5: Why did you not use plasma treated water in the recipe? The method you used will not find the practical application in the industry.
Answer: This section was corrected. The translation was wrong and incomplete.
- Line 185-186: The sentence needs to rephrased.
Answer: it’s corrected.
- Point 2.13: What did you examine: sensory quality or acceptability? The more detailed information on the panel training is needed in case of sensory quality.
Answer: it’s corrected.
- Point 2.14: The statistical analysis description needs to be elaborated.
Answer: it’s corrected.
- Line 227: Full stop after “weight” is not needed.
Answer: it’s corrected.
- Line 229 and 231: Please use the proper way to present the units.
Answer: it’s corrected.
- Line 234 and elsewhere: Please use the same size of the font.
Answer: it’s corrected.
- Lines 316-317: What does it have to do with your research?
Answer: it’s removed.
- Line 335-339: Was the research material prepared similarly to yours? If so please be more precise.
Answer: it’s revised.
- Line 361: What was the control sample?
Answer: it’s corrected: sample G (100% nitrate-without PAW treatment)
- General remarks: The influence of a rosemary extract on the quality of meat products is well known. What is interesting is comparison of samples with added plasma activated water and those without this addition. However, you have resigned from the comparison after microbiological tests. Therefore the precise aim of the research is needed to assess its soundness.
Answer: In this regard, sample B as a sample treated with PAW (containing 100% rosemary extract) and sample G as a control (without treatment with PAW-100% nitrite), was select to investigate and compare the simultaneous effect of PAW and rosemary extract on the physicochemical characteristics of sausage samples during storage.

Round 2
Reviewer 1 Report
Comments and Suggestions for Authors
The authors made most of the requested changes and the article improved considerably.
Author Response
Dear Reviewer,
Thank you for your valuable comments on our work. Your insights and suggestions have greatly contributed to the improvement of our manuscript, ultimately helping us achieve a higher level of quality. We sincerely appreciate your time and expertise.
Also, the English writing of the manuscript improved.
Best wishes
Reviewer 2 Report
Comments and Suggestions for Authors
The authors have improved the manuscript. However, it still needs some corrections:
1. The authors have not answered my question about the lack of calibration curve in their TPC measurement.
2. Line 245: The units are different to those presented in the Results section
3. Paragraph 3.7: The sample coding should be the same in the text and the table. Please include samples' codes in the table in brackets.
4. My doubt about not using PAW in the sausage formula remains unanswered. Form practical point of view, soaking slices of sausage in PAW is pointless. How to use this procedure in commercial settings?
Table 5: optimized not optimize
Comments on the Quality of English LanguageThere are minor errors to be corrected.
Author Response
Dear Reviewer,
We would like to express our gratitude for your valuable comments, which helped us qualify for a higher level. Your guidance and suggestions have been instrumental in enhancing the quality of our work, and we truly appreciate your expertise and input. Thank you once again for your invaluable contribution.
- The authors have not answered my question about the lack of calibration curve in their TPC measurement.
Answer: It is added to the manuscript.
- Line 245: The units are different to those presented in the Results section
Answer: it’s corrected. For TPC is mg gallic acid/g dr.wt and for TFC is mg QE/ g dr.wt.
- Paragraph 3.7: The sample coding should be the same in the text and the table. Please include samples' codes in the table in brackets.
Answer: it’s corrected.
- My doubt about not using PAW in the sausage formula remains unanswered. Form practical point of view, soaking slices of sausage in PAW is pointless. How to use this procedure in commercial settings?
Answer: You are absolutely right and this study definitely has many flaws that we will try to fix in the next studies. But our purpose of conducting this study was to investigate whether the activated water plasma has the potential to reduce the microbial load with the least effect on the physicochemical characteristics of sausages, as a complementary method after heat treatment (pasteurization). Similar to our work, Wang et al. (2021) also used activated plasma water after preparing and cooking chicken breasts to reduce its microbial load( Wang, J.; Han, R.; Liao, X.; Ding, T. Application of plasma-activated water (PAW) for mitigating methicillin-resistant Staphylococcus aureus (MRSA) on cooked chicken surface. LWT 2021, 137, 110465).
According to the studies, the water that is treated and activated with cold plasma contains compounds (such as ROS-RON reactive species) with antimicrobial properties and this water has the potential to reduce the microbial load.
Table 5: optimized not optimize
Answer: it’s corrected.
